# Self-Supervised Pseudodata Filtering for Improved Replay with Sub-Optimal Generators

## Abstract

Continual learning on a sequence of tasks without forgetting previously acquired knowledge is one of the main challenges faced by modern deep neural networks. In the class-incremental scenario, one of the most difficult continual learning problems, new classes are presented to a classifier over time. The model needs to be able to learn and recognize these new classes while also retaining its knowledge of previously witnessed ones. To achieve this, the model has to revisit previous classes in some form, either by analysing stored exemplars or by using artificially generated samples. The latter approach, Generative Replay, usually relies on a separate generator trained alongside the main classifier. Since the generator also needs to learn continually, it is retrained on every task, using its own generated samples as training data representing older classes. This can lead to error propagation and accumulating features unimportant or confusing for the classifier, reducing the overall performance for larger numbers of tasks. We propose a simple filtering mechanism for mitigating this issue – whenever pseudodata is generated for a new task, the classifier can reject samples it is not able to classify with sufficient confidence, thus preventing itself from retraining on poor-quality data. We tested this mechanism using combinations of Bayesian neural classifiers and two different generators: a Variational Autoencoder and Real-value Non-Volume Preserving Normalizing Flow. We show that the improvement in the classification accuracy grows with the number of tasks, suggesting this approach is particularly useful for the most challenging continual learning scenarios, where very many tasks are learned in a sequence.

## 1 Introduction

Catastrophic forgetting of previously learned knowledge after being trained on a new task is one of the main drawbacks of modern deep neural networks (French (1999); Jedlicka et al. (2022)). The ability to mitigate this issue, and learn continually, is crucial in many realistic machine learning applications, including autonomous machines navigating in changing environments and real-time decision makers having to adapt and react to shifting incoming data distributions (Shaheen et al. (2022)). In classification problems, such continual learning scenarios are often labeled as Task-, Domain- or Class-Incremental Learning (IL) (Van de Ven & Tolias (2019)). These scenarios differ mostly in terms of the availability of the task identity: In a Task-IL scenario, the model is aware of which task it's solving both in the training and the prediction phase, a Domain-IL model knows the task identity only during training, while a Class-IL model lacks such knowledge altogether. These scenarios are further explained in figure 1.

While challenging for artificial neural networks, catastrophic forgetting does not affect biological learning agents, such as humans and other mammals. The way we interact with our environment is inherently time-dependent – we learn new patterns and skills sequentially, building upon and expanding the previously acquired knowledge instead of completely overwriting it. Several classes of mechanisms are responsible for this evolutionary success, but the one most relevant in the context of this work, is experience replay (Abraham (2008); Yger & Gilson (2015); McClelland et al. (1995); Rasch & Born (2013)).

To stabilize the previously learned patterns, an artificial neural network can revisit old experiences, in the mechanism called "replay" or "rehearsal. In the mammalian brain, such reminiscence is ob-

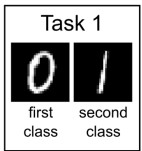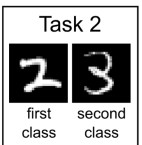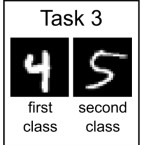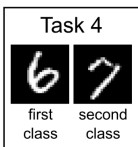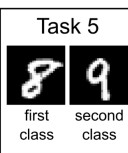

Figure 1: SplitMNIST task protocol. In task-incremental scenarios the model learns classes pairwise and during testing it knows which pair the current image belongs to. In domain-incremental scenario the model needs to decide whether the image belongs to the first or the second class in its corresponding pair, but the identity of the pair is irrelevant (e.g., all odd numbers in MNIST get the same label assigned). In class-incremental scenario the model needs to learn how to distinguish between a given digit and all other digits witnessed so far. Figure adapted from Van de Ven & Tolias (2019).

served for example during sleep, when the hippocampal activity reinstates activity in the neocortical processing systems. One hypothesis regarding this behaviour is that it is responsible for effective consolidation and stabilization of long-term memories (McClelland et al. (1995)). The simplest form of rehearsal would be to store a subset of previously encountered training data and iteratively retrain the model from scratch every time a new task arises. However, storing exact copies of past experiences would be impossible in capacity-constrained animal brains, deeming such an approach not biologically plausible. In machine learning there are situations when data storage becomes impractical or impossible, for example, due to privacy issues or computational constraints. Instead, a growing number of methods rely on generative replay, where the data distribution is learned by a generative model. By sampling from the generator, it is possible to access features relevant to the previous tasks and interleave them with the current dataset. A basic architecture of a generative replay framework, where the generator and the solver are separate neural models, was proposed by Shin et al. (2017), and the concept was further developed by Van de Ven et al. (2020) and Kirichenko et al. (2021), to name a few.

In this work we focus on Shin et al.'s dual-model architecture, even though it does not achieve the highest performance on standard benchmarks (Van de Ven et al., 2020; Kirichenko et al., 2021). We make this choice for two main reasons. First, the dual-model architecture can be applied to any neural classifier without additional modifications to the network's structure. This flexibility makes it convenient in situations when classifier (or, more generally, task solver) models are already well-established and trained, and the requirement to learn class-incrementally arises as an additional functionality, without being considered during the model's design. In such cases, the implementation of suitable generators eliminates the need for a complete redesign and retraining of the classifier, such as incorporating feedback connections. A second noteworthy advantage of the dual-model approach lies in its simplicity. The process of generating the pseudodata (for example, images belonging to the previously learned classes) and training the classifier can be clearly separated, facilitating a more transparent understanding of each component's contribution to the overall performance.

We propose a simple and universal mechanism for improving generative replay models, addressing one of their common weaknesses - poor scalability to a larger number of tasks due to error propagation in the generator (Lesort et al. (2019a); Aljundi et al. (2019)). As we investigate a scenario when the original training data cannot be stored, the generative model also needs to learn continually, iteratively retraining itself on its own generated samples. If pseudodata generated for one of the tasks contains features unnecessary or confusing for the classifier, there exists a chance that these features are going to be preserved in the distribution learned by the generator, detrimentally affecting replay's effectiveness for all the subsequent tasks. To combat this, we propose a method of filtering the generated data by allowing the classifier to automatically select best-quality samples and remove data lacking necessary features — in other words, we allow the solver to self-supervise the replay process. A conceptually similar approach, using the classifier's confidence about the generated samples as a contribution to the generator's loss function, was explored by Aljundi et al. (2019), but the policy we propose can be treated as a stricter variant, when pseudodata quality is assessed on the dataset level instead of just encouraging the model to improve it with time.

We show that when the number of tasks is sufficiently high, self-supervised filtering of pseudodata has a small, but positive effect on the performance in terms of accuracy, and that its contribution is

strongly, positively correlated with the number of tasks. Notably, this observation is general enough that the proposed method can be applied to most of the existing generative replay architectures.

To sum up, the main contribution of our paper is a general technique of filtering pseudodata, improving the performance of generative replay in class-incremental learning scenarios. We also investigate the scalability of this technique with the number of tasks, an approach that can be helpful for the community working on the catastrophic forgetting problem.

## 2 METHODS

In this section, we describe the models we used for experiments, the dataset, and the training procedure applied. The code is publicly available here: *link to code repository anonymized for peer review*

As mentioned, the main contribution of our work is a method of filtering pseudodata sampled from the generator. In order to do this we label each generated image using the classifier and then remove samples classified with confidence below a selected threshold. Here by "confidence" we mean the highest value returned by the softmax function in the output layer. The higher the threshold, the stricter the filtering policy.

### 2.1 MODELS USED IN THE EXPERIMENTS

To investigate and demonstrate the effectiveness of the proposed filtering procedure we performed classification experiments using various neural network models. To generate pseudodata we used a Real-valued Non-Volume Preserving (RealNVP/RNVP) Normalizing Flow or a Variational Autoencoder (VAE). To classify the data we trained a standard, densely connected Bayesian Neural Network (BNN) and its regularized variant following the method of Variational Continual Learning (VCL). The models, described in detail below, were combined into four experimental setups: RNVP+BNN, RNVP+VCL, VAE+BNN and VAE+VCL.

#### 2.1.1 BAYESIAN NEURAL NETWORKS AND VARIATIONAL CONTINUAL LEARNING

Bayesian neural networks are models that incorporate Bayesian methods to model uncertainty in the network's weights and biases (Jospin et al. (2022)). Parameters in a BNN are given as probability distributions instead of point values, so the exact strength of connections is drawn every time a forward pass is performed. As such, if run several times with the same input, a BNN generates a vector of predictions, the variance of which can be used to reason about the result's uncertainty. This is particularly useful in tasks where uncertainty plays a critical role, such as medical diagnosis or financial risk assessment. Reducing catastrophic forgetting in this class of models is thus crucial for enabling continual learning in such areas. Moreover, we believe that densely-connected BNNs are sufficiently similar to traditional, non-Bayesian networks, for our method to work in both cases.

As a stochastic model, BNNs need to be trained using probabilistic inference. The most common families of methods used for this purpose are various forms of Markov-Chain Monte Carlo (MCMC) sampling and Variational Inference (VI). In this work we used the latter to train the classifiers, as it scales better to larger models.

Variational Inference approximates the true posterior distribution of the weights and biases of the neural network with a simpler, parametric distribution. The parameters of the approximate distribution are learned using gradient descent to minimize the Kullback-Leibler (KL) divergence. However, since direct calculation of KL would require knowledge about the form of the true posterior, the optimization is performed indirectly by maximizing the evidence lower bound (ELBO):

$$\mathcal{L}(\theta, \phi; \mathbf{X}, \mathbf{y}) = \mathbb{E}q_\phi(\theta)[\log p_\theta(\mathbf{y}|\mathbf{X}, \theta)] - \text{KL}(q_\phi(\theta)||p(\theta)) \tag{1}$$

where $\mathbf{X}$ and $\mathbf{y}$ are the input data and output labels, respectively, $\theta$ and $\phi$ are the parameters of the neural network and the approximate distribution, and $p$ is the prior distribution over the weights. The first term of ELBO is the expected log-likelihood of the data under the current approximate distribution (implemented as categorical cross entropy in our case), and the second term is the KL divergence between the approximate distribution and the prior.

Due to high computational requirements of calculating and storing dependencies between the network's parameters, usually both the approximate posterior and the prior take the form of a fully-factorized Gaussian (Izmailov et al. (2021)).

In Bayesian modeling, the prior distribution represents the uncertainty about the parameters before any data is observed. In standard BNN training, it usually takes the form of a unit normal distribution, as it's difficult to form any assumptions about the network's parameters before training. However, such an assumption becomes easier to make when a new task appears in a continual learning scenario – the prior knowledge in this case corresponds to the knowledge the network acquired while learning the previous tasks. In other words, to utilize this information, the model trained on task *n* should be used as a prior for task *n+1*. This approach, Variational Continual Learning was first proposed by Nguyen et al. (2017), and can be treated as a regularization-based method of continual learning for BNNs.

In the brain, such regularization, called "metaplasticity" allows memories to remain stable by reducing the plasticity of synaptic connections as new memories are formed (Abraham (2008)). In VCL, encouraging the modification of the parameters that carry the least information regarding the previous task is handled automatically thanks to the KL divergence term. Metaplasticity-based regularization is a base of several state-of-the-art methods for continual learning, such as Synaptic Intelligence (Zenke et al. (2017)) and Elastic Weight Consolidation (Kirkpatrick et al. (2017)), but, on their own, such methods fail in class-IL scenarios, where using replay-based algorithms seems to be necessary (Lesort et al. (2019b); Van de Ven et al. (2020)). Here we used VCL to regularize weight updates in selected models, thus introducing metaplasticity, to check how it contributes to the BNN's performance.

The classifier used for the experiments presented in this paper was a densely connected BNN with two hidden layers, 256 nodes in each layer. It is further referred to as a "BNN" or a "VCL" model, depending on whether a standard unit Gaussian or a previous posterior was used as a prior distribution for each task.

### 2.1.2 REALNVP NORMALIZING FLOWS

One of the generators used in the experiments is a Real-valued Non-Volume Preserving Normalizing Flow, proposed by Dinh et al. (2016). Normalizing Flows (NFs) are a class of generative models that aim to model complex probability distributions by performing a series of invertible transformations on a simple base (latent) distribution. In one variant of such a transformation, RealNVP, the input (a sample from the base distribution) is divided into two subsets – the values in the first one are scaled and shifted by factors calculated based on the second one. While this transformation as a whole needs to be invertible, the function used to calculate the factors does not, which allows the use of even complex neural models to accurately capture the dependencies between variables. In practice, a RealNVP generator is built using many blocks of such transformations (coupling layers). The input subsets switch places after each layer so that each value in the input tensor undergoes a transformation once every second layer. Since NFs are invertible, it is possible to calculate the exact likelihood of training data and minimize it directly with gradient descent optimizers.

The generator used in this work, further referred to as the "RealNVP" model, consisted of eight blocks, each block containing a permutation layer (switching the input subsets) and a RealNVP transformation layer, followed by batch normalization. To calculate scale and shift factors, each RNVP coupling layer used a multilayer perceptron with two hidden layers, 56 nodes per layer. The model was implemented as a conditional generator which means that while the parameters of the Flow were shared between all the classes in the training dataset, each class was represented by a separate latent distribution (a multivariate Gaussian). As a result, the samples to be transformed by the coupling layers were drawn from different distributions, depending on which class was chosen for generation.

### 2.1.3 VARIATIONAL AUTOENCODERS

Variational Autoencoders (Kingma & Welling (2013)) are a well-established class of probabilistic generators where each model consists of two neural networks: an encoder and a decoder. The encoder maps input data to a latent distribution (usually a multivariate Gaussian) while the decoder reconstructs the output using a sample from the said distribution. As opposed to Normalizing Flows,

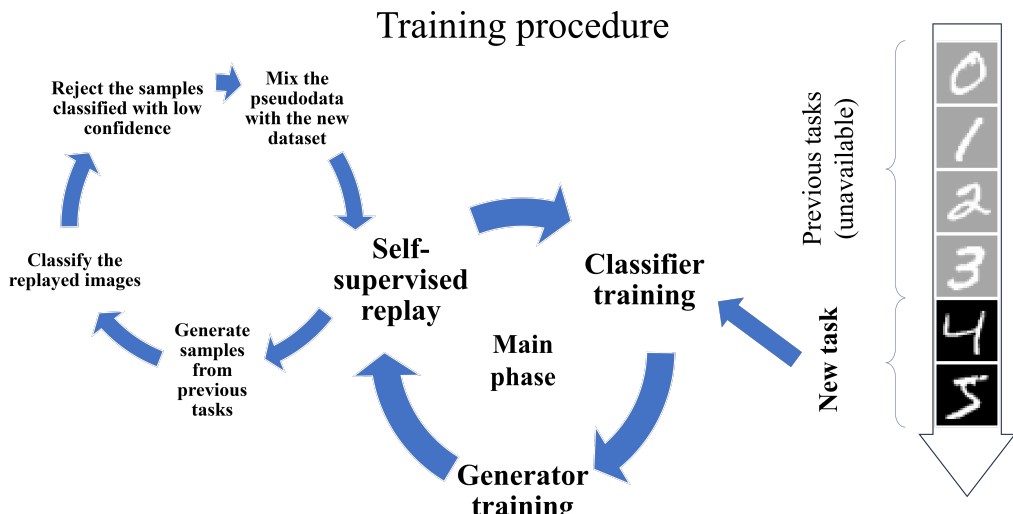

Figure 2: Schematic view of the training procedure. In the class-incremental learning scenario, classes are presented sequentially, in exclusive groups, but the model needs to keep the ability to recognize all classes witnessed so far. After training the solver and the generator on all data available, the framework enters the self-supervised stage when it generates and filters pseudodata used for the next iteration of training.

Variational Autoencoders are not invertible and as such need to be trained indirectly by maximizing the evidence lower bound.

The VAE used in this work was implemented as a convolutional, non-conditional generator – a single, two-dimensional latent distribution was shared between classes.

## 2.2 EXPERIMENTAL PROCEDURE

We formulated the learning problem as a class-incremental scenario. During each task, the model was presented with only two classes of images, but it was expected to be able to classify all classes witnessed so far. Figure 2 contains a schematic depiction of the training procedure.

### 2.2.1 EMNIST DATASET

To extend the number of tasks beyond the maximum five provided by MNIST dataset, a standard benchmark in the field (LeCun (1998); Parisi et al. (2019)), we chose to use EMNIST Balanced (Cohen et al. (2017)), which serves as an extension of the former. It contains pictures of both digits and letters, 47 classes in total. Here we report results of training on up to 16 tasks (covering 32 classes), since for longer training protocols the quality of the generators would often decrease to a point where they do not generate enough good-quality samples to be accepted by the classifier, especially with stricter filtering.

For both training and evaluation, we scaled the pixel values to the range [0, 1]. For experiments using RealNVP we applied additional preprocessing converting pixel intensities to logits as recommended by Dinh et al. (2016).

### 2.2.2 MODEL TRAINING AND PSEUDODATA GENERATION

The experimental procedure consisted of three main components: model training, pseudodata generation, and testing. First the main component, described in algorithm 1, combined the incoming real data with generated pseudodata and trained both classifier and the generator on all the available images. An internal loop was used for pseudodata generation (algorithm 2). There, the current state of the generator was used to sample a fixed number of images, so that the training dataset consisting

of real and pseudodata was class-balanced. Next, these images were classified by the solver and all samples classified below the assigned level of confidence were removed — a step that we refer to as "pseudodata-filtering". This filtering was repeated until the pseudo-dataset reached the requested size, chosen to be 2500 per class in our implementation. Finally, after training on each task, the model was asked to classify real test images belonging to all previously observed classes, without knowing when a particular class was encountered. In the next section, we report the results in terms of accuracy, averaged over all the random initializations of the models'parameters and sampling functions.

---

**Algorithm 1** Incremental Learning Procedure

---

 1: Initialize a generator
 2: **for** each task **do**
 3:     Load and preprocess the next dataset
 4:     **if** task_id $> 0$ **then**
 5:         Generate pseudodata, a fixed number of images for each class
 6:     **end if**
 7:     Concatenate the real and pseudo datasets
 8:     Permute the training dataset
 9:     Initialize classifier with suitable prior
10:     Train the classifier
11:     Train the generator
12: **end for**

---

---

**Algorithm 2** pseudodata Generation

---

 1: current_pseudo_size $\leftarrow 0$
 2: **while** current_pseudo_size $<$ pseudo_dataset_size **do**
 3:     Generate samples
 4:     Classify the generated images $n$ times
 5:     Calculate the mean prediction (confidence) for each class
 6:     Remove samples classified below the confidence threshold
 7:     **if** conditional generator **then**
 8:         Remove samples classified incorrectly
 9:     **end if**
10:     Add remaining samples to the pseudo-dataset
11:     Increase current_pseudo_size by the number of accepted samples
12: **end while**

---

## 3 RESULTS

We performed all the experiments thirty times, with different random seeds. The models were tested after training on each task by classifying test data belonging to all the classes witnessed so far. Whenever filtering was applied, the confidence threshold was set to 90, 95, or 99 percent. Especially with higher thresholds, some generators entered infinite loops at various later points during training, when they kept trying to generate replay samples that kept being rejected by the classifier. In such circumstances, the training was terminated, so not all thirty resulting data points are available for higher task numbers. We compared the performance of models trained with and without filtering using Student's T-test for the difference of means (figure 3) and Mood's test for the difference of medians (figure 4. The exact p-values, as well as the results of the Mann-Whitney U test, for comparison, are provided in the Appendix.

Overall, models trained with pseudodata filtering performed better in terms of mean accuracy in 75% of cases. Moreover, this effect scales with the number of tasks as seen in Table 1 showing Pearson's correlation coefficients between the improvement in accuracy and the length of training, the high positive values suggest a dependency between them. For the difference of medians we observed a positive trend, albeit less consistent and not statistically significant given the chosen thresholds ($\alpha = 0.05$ or $\alpha = 0.01$), suggesting a higher data granularity required for fair comparison with this

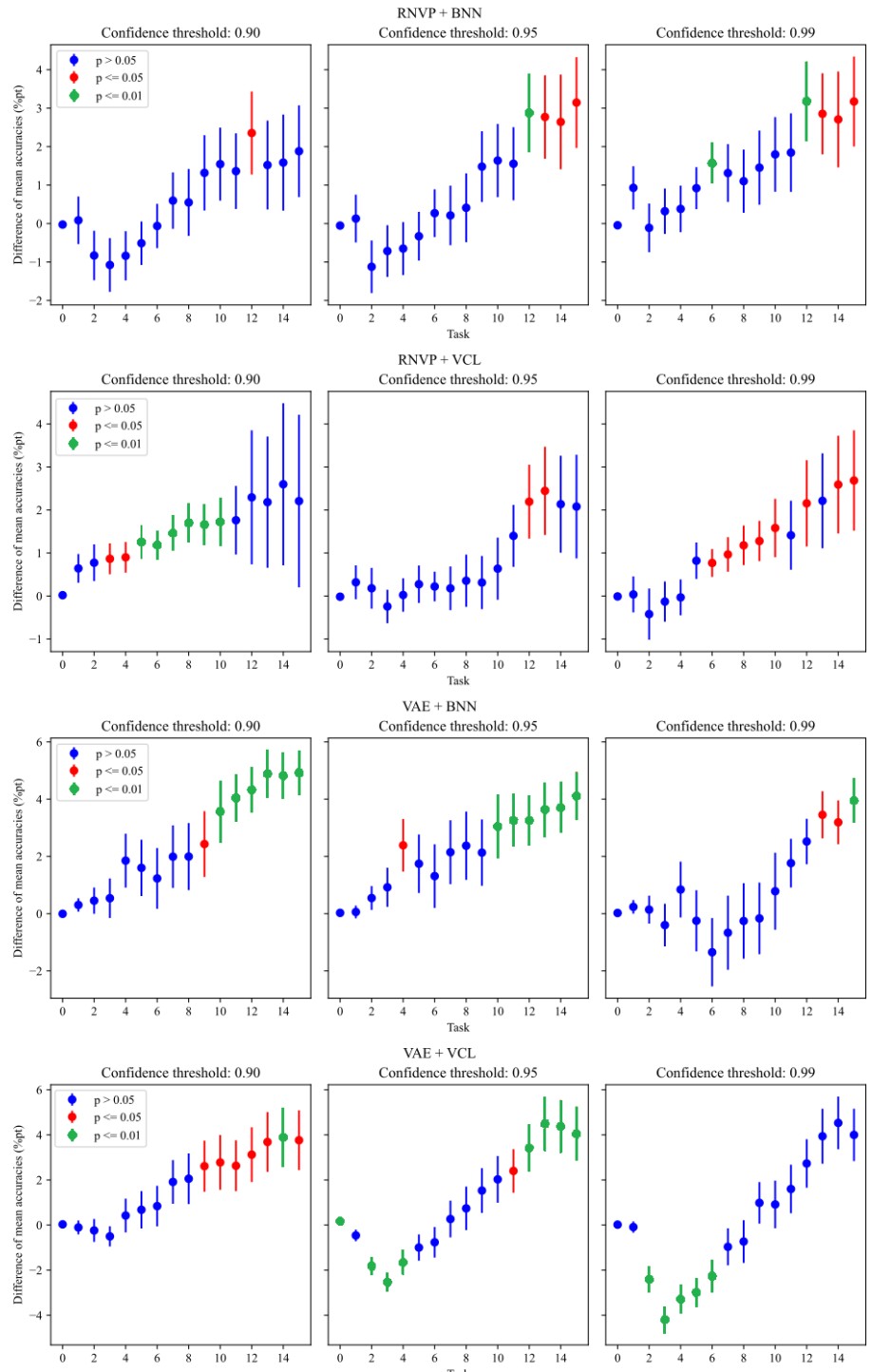

Figure 3: Differences between mean accuracies of models trained with and without filtering. Results for which the Student's t-test returned p value not larger than 0.05 or 0.01 are marked with red and green, respectively. Error bars represent standard errors of the difference of means.

statistic. Another issue visible in the figures, especially with VAE as a generator, is that the filtering procedure had a negligible or even detrimental effect when the number of tasks was low. We suggest an interpretation of this phenomenon and elaborate on its consequences for the applicability of our method in the Discussion.

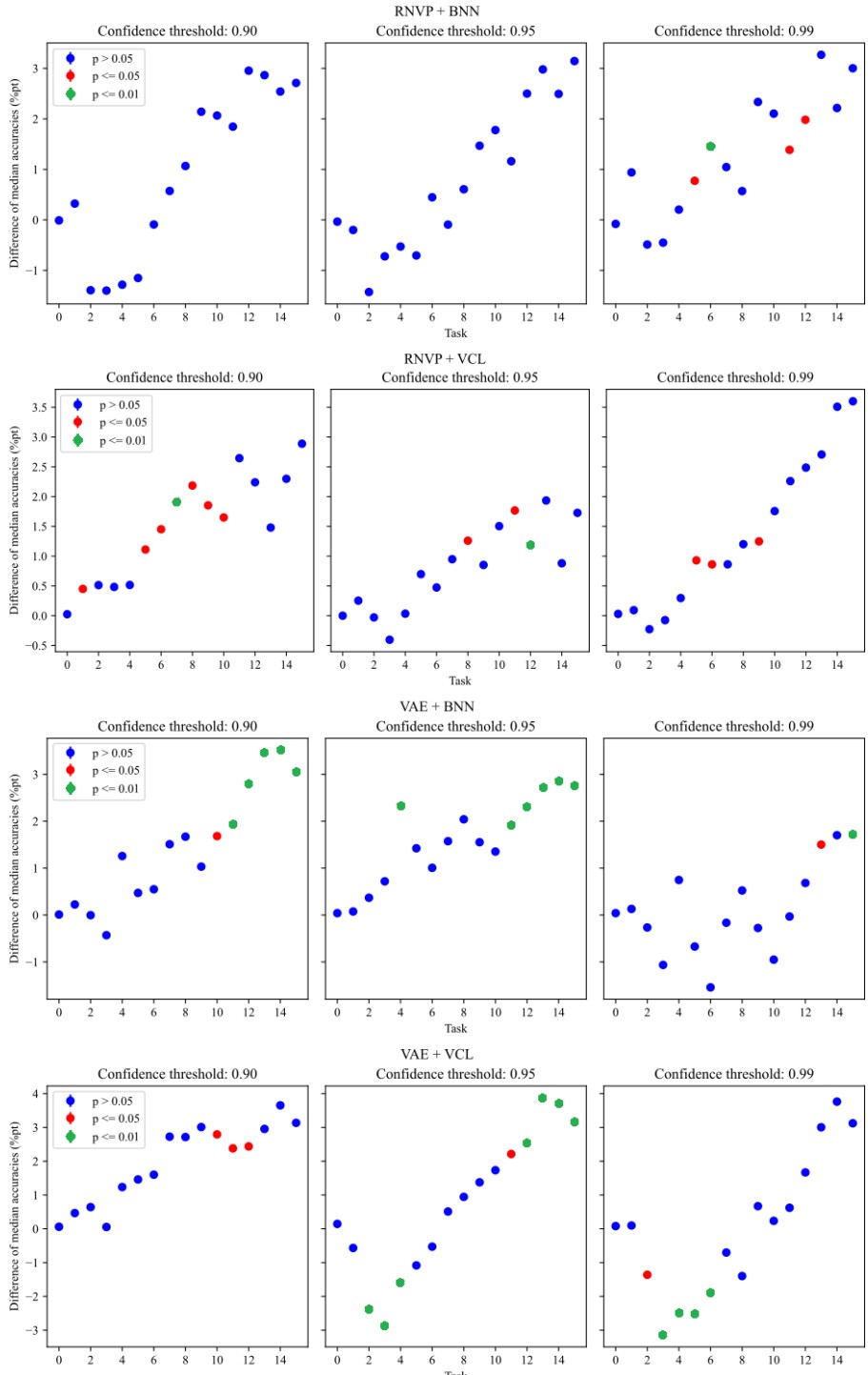

Figure 4: Differences between median accuracies of models trained with and without filtering. Results for which the Mood's test returned p value not larger than 0.05 or 0.01 are marked with red and green, respectively.

| Model | Threshold | R |
|---|---|---|
| RNVP+BNN | 0.90 | 0.87 |
| | 0.95 | 0.92 |
| | 0.99 | 0.93 |
| RNVP+VCL | 0.90 | 0.97 |
| | 0.95 | 0.86 |
| | 0.99 | 0.97 |
| VAE+BNN | 0.90 | 0.97 |
| | 0.95 | 0.96 |
| | 0.99 | 0.76 |
| VAE+VCL | 0.90 | 0.97 |
| | 0.95 | 0.89 |
| | 0.99 | 0.80 |

Table 1: Pearson's correlation coefficients between differences of mean accuracies between models with and without filtering, and the number of tasks. All results are statistically significant with $\alpha = 0.05$.

## 4 DISCUSSION

In this paper, we presented a method of filtering samples from a generative model used for data replay in class-incremental continual learning scenario. Our original hypothesis consisted of two parts: first, that data filtering will improve the accuracy of a classifier trained with generative replay; second, that this improvement will positively scale with the number of tasks. The justification behind the first part is that by allowing the solver to select data it can classify with the highest level of confidence, we automatically reinforce the presence of features important for distinguishing between classes in the replayed dataset. As for scaling of the effect, we assume that without data filtering more errors can propagate from task to task, since the generator may learn to repeat its own mistakes. With filtering, if such a mistake would reduce the sample's usefulness for learning the task, it will be removed from the training set used both by the solver and the generator.

The results we present confirm both parts of our hypothesis to some degree. First of all, in the majority of cases where performance with and without filtering was significantly different, the filtering did result in improved accuracy, except for the cases when the number of tasks was small and/or the confidence threshold was very high. The reason for this may be that for the first few tasks, the error propagation in the generator is not very significant, and radical filtering of the pseudodata reduces the diversity of samples, limiting the solver's ability to generalize. This suggests that the confidence threshold is a hyperparameter that is very important to optimize while taking into consideration the expected scale of the learning problem. However, even when the initial improvement was negligible or negative at the beginning, it grew as the training progressed, eventually reaching positive values in all investigated model configurations. As the difference in accuracy and the number of tasks seem to be strongly correlated, this pattern can be expected to be relevant for even larger problems, and the filtering can be even more useful when even more tasks need to be learned in a sequence. Future research should focus on exploring other filters than the classifier's confidence and assessing the technique's robustness for more complex datasets.

When it comes to related work, a conceptually similar approach was proposed by Aljundi et al. (2019) who control sampling of memories in exact and generative replay. The major difference is that they realize this by modifying the loss function of the generator, for example penalizing the entropy of classifier-assigned labels, encouraging the generator to create data that the classifier will be confident about. As such, this model-focused approach differs from our data-focused one, but in their ablation studies, Aljundi et al. (2019) show that this entropy term is essential to outperform the baseline, which works in favor of using classifier's results for selective generative replay.

In summary, the self-supervised pseudodata filtering can be a useful technique for improving generative replay when the number of tasks is large. Being a general method, it can become a helpful addition other approaches combating catastrophic forgetting in deep neural networks.

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

# A  RESULTS OF STATISTICAL TESTS

## A.1  STUDENT'S T-TEST

| Confidence threshold | Trained task | Accuracy difference | p |
|---|---|---|---|
| 0.900 | 0 | -0.006 | 0.831 |
| 0.900 | 1 | 0.303 | 0.206 |
| 0.900 | 2 | 0.453 | 0.334 |
| 0.900 | 3 | 0.539 | 0.445 |
| 0.900 | 4 | 1.852 | 0.057 |
| 0.900 | 5 | 1.600 | 0.115 |
| 0.900 | 6 | 1.231 | 0.259 |
| 0.900 | 7 | 1.990 | 0.079 |
| 0.900 | 8 | 1.994 | 0.098 |
| 0.900 | 9 | 2.431 | 0.042 |
| 0.900 | 10 | 3.564 | 0.002 |
| 0.900 | 11 | 4.040 | 0.000 |
| 0.900 | 12 | 4.323 | 0.000 |
| 0.900 | 13 | 4.884 | 0.000 |
| 0.900 | 14 | 4.816 | 0.000 |
| 0.900 | 15 | 4.928 | 0.000 |
| 0.950 | 0 | 0.027 | 0.270 |
| 0.950 | 1 | 0.059 | 0.796 |
| 0.950 | 2 | 0.545 | 0.204 |
| 0.950 | 3 | 0.921 | 0.191 |
| 0.950 | 4 | 2.388 | 0.013 |
| 0.950 | 5 | 1.744 | 0.098 |
| 0.950 | 6 | 1.312 | 0.250 |
| 0.950 | 7 | 2.145 | 0.064 |
| 0.950 | 8 | 2.373 | 0.056 |
| 0.950 | 9 | 2.132 | 0.077 |
| 0.950 | 10 | 3.047 | 0.010 |
| 0.950 | 11 | 3.258 | 0.001 |

| Confidence threshold | Trained task | Accuracy difference | p |
|---|---|---|---|
| 0.950 | 12 | 3.257 | 0.001 |
| 0.950 | 13 | 3.636 | 0.001 |
| 0.950 | 14 | 3.708 | 0.000 |
| 0.950 | 15 | 4.113 | 0.000 |
| 0.990 | 0 | 0.024 | 0.400 |
| 0.990 | 1 | 0.236 | 0.332 |
| 0.990 | 2 | 0.140 | 0.778 |
| 0.990 | 3 | -0.400 | 0.599 |
| 0.990 | 4 | 0.842 | 0.400 |
| 0.990 | 5 | -0.249 | 0.819 |
| 0.990 | 6 | -1.348 | 0.269 |
| 0.990 | 7 | -0.666 | 0.612 |
| 0.990 | 8 | -0.255 | 0.849 |
| 0.990 | 9 | -0.165 | 0.897 |
| 0.990 | 10 | 0.783 | 0.647 |
| 0.990 | 11 | 1.764 | 0.254 |
| 0.990 | 12 | 2.520 | 0.086 |
| 0.990 | 13 | 3.453 | 0.030 |
| 0.990 | 14 | 3.191 | 0.033 |
| 0.990 | 15 | 3.951 | 0.008 |

Table 2: Results of Student's T-test for VAE+BNN model configuration.

| Confidence threshold | Trained task | Accuracy difference | p |
|---|---|---|---|
| 0.900 | 0 | 0.030 | 0.649 |
| 0.900 | 1 | -0.107 | 0.727 |
| 0.900 | 2 | -0.240 | 0.639 |
| 0.900 | 3 | -0.503 | 0.263 |
| 0.900 | 4 | 0.422 | 0.575 |
| 0.900 | 5 | 0.678 | 0.420 |
| 0.900 | 6 | 0.839 | 0.365 |
| 0.900 | 7 | 1.912 | 0.063 |
| 0.900 | 8 | 2.054 | 0.087 |
| 0.900 | 9 | 2.611 | 0.033 |
| 0.900 | 10 | 2.777 | 0.036 |
| 0.900 | 11 | 2.629 | 0.033 |
| 0.900 | 12 | 3.122 | 0.020 |
| 0.900 | 13 | 3.684 | 0.013 |
| 0.900 | 14 | 3.888 | 0.007 |
| 0.900 | 15 | 3.762 | 0.010 |
| 0.950 | 0 | 0.170 | 0.001 |
| 0.950 | 1 | -0.460 | 0.093 |
| 0.950 | 2 | -1.810 | 0.000 |
| 0.950 | 3 | -2.529 | 0.000 |
| 0.950 | 4 | -1.653 | 0.007 |
| 0.950 | 5 | -0.999 | 0.102 |
| 0.950 | 6 | -0.765 | 0.277 |
| 0.950 | 7 | 0.267 | 0.751 |
| 0.950 | 8 | 0.740 | 0.459 |
| 0.950 | 9 | 1.529 | 0.142 |
| 0.950 | 10 | 2.026 | 0.070 |
| 0.950 | 11 | 2.399 | 0.023 |
| 0.950 | 12 | 3.422 | 0.004 |

| Confidence threshold | Trained task | Accuracy difference | p |
|---:|---:|---:|---:|
| 0.950 | 13 | 4.488 | 0.001 |
| 0.950 | 14 | 4.380 | 0.001 |
| 0.950 | 15 | 4.056 | 0.002 |
| 0.990 | 0 | 0.018 | 0.786 |
| 0.990 | 1 | -0.090 | 0.723 |
| 0.990 | 2 | -2.402 | 0.000 |
| 0.990 | 3 | -4.201 | 0.000 |
| 0.990 | 4 | -3.287 | 0.000 |
| 0.990 | 5 | -2.986 | 0.000 |
| 0.990 | 6 | -2.263 | 0.005 |
| 0.990 | 7 | -0.966 | 0.274 |
| 0.990 | 8 | -0.732 | 0.471 |
| 0.990 | 9 | 0.983 | 0.323 |
| 0.990 | 10 | 0.913 | 0.539 |
| 0.990 | 11 | 1.599 | 0.376 |
| 0.990 | 12 | 2.730 | 0.272 |
| 0.990 | 13 | 3.939 | 0.167 |
| 0.990 | 14 | 4.530 | 0.095 |
| 0.990 | 15 | 3.998 | 0.205 |

Table 3: Results of Student's T-test for VAE+VCL model configuration.

| Confidence threshold | Trained task | Accuracy difference | p |
|---:|---:|---:|---:|
| 0.900 | 0 | -0.026 | 0.502 |
| 0.900 | 1 | 0.085 | 0.894 |
| 0.900 | 2 | -0.831 | 0.213 |
| 0.900 | 3 | -1.077 | 0.139 |
| 0.900 | 4 | -0.838 | 0.207 |
| 0.900 | 5 | -0.512 | 0.381 |
| 0.900 | 6 | -0.064 | 0.915 |
| 0.900 | 7 | 0.596 | 0.432 |
| 0.900 | 8 | 0.549 | 0.541 |
| 0.900 | 9 | 1.318 | 0.196 |
| 0.900 | 10 | 1.545 | 0.119 |
| 0.900 | 11 | 1.362 | 0.183 |
| 0.900 | 12 | 2.354 | 0.039 |
| 0.900 | 13 | 1.521 | 0.205 |
| 0.900 | 14 | 1.585 | 0.222 |
| 0.900 | 15 | 1.880 | 0.132 |
| 0.950 | 0 | -0.053 | 0.134 |
| 0.950 | 1 | 0.129 | 0.840 |
| 0.950 | 2 | -1.124 | 0.116 |
| 0.950 | 3 | -0.716 | 0.304 |
| 0.950 | 4 | -0.651 | 0.361 |
| 0.950 | 5 | -0.330 | 0.614 |
| 0.950 | 6 | 0.270 | 0.674 |
| 0.950 | 7 | 0.211 | 0.791 |
| 0.950 | 8 | 0.410 | 0.657 |
| 0.950 | 9 | 1.480 | 0.123 |
| 0.950 | 10 | 1.637 | 0.100 |
| 0.950 | 11 | 1.554 | 0.118 |
| 0.950 | 12 | 2.877 | 0.009 |
| 0.950 | 13 | 2.770 | 0.017 |

Continued on next page

| Confidence threshold | Trained task | Accuracy difference | p |
|---|---|---|---|
| 0.950 | 14 | 2.642 | 0.042 |
| 0.950 | 15 | 3.145 | 0.013 |
| 0.990 | 0 | -0.045 | 0.259 |
| 0.990 | 1 | 0.929 | 0.114 |
| 0.990 | 2 | -0.112 | 0.863 |
| 0.990 | 3 | 0.320 | 0.599 |
| 0.990 | 4 | 0.381 | 0.542 |
| 0.990 | 5 | 0.920 | 0.107 |
| 0.990 | 6 | 1.572 | 0.006 |
| 0.990 | 7 | 1.313 | 0.095 |
| 0.990 | 8 | 1.102 | 0.198 |
| 0.990 | 9 | 1.453 | 0.149 |
| 0.990 | 10 | 1.797 | 0.078 |
| 0.990 | 11 | 1.844 | 0.085 |
| 0.990 | 12 | 3.180 | 0.005 |
| 0.990 | 13 | 2.852 | 0.012 |
| 0.990 | 14 | 2.707 | 0.041 |
| 0.990 | 15 | 3.173 | 0.012 |

Table 4: Results of Student's T-test for RNVP+BNN model configuration.

| Confidence threshold | Trained task | Accuracy difference | p |
|---|---|---|---|
| 0.900 | 0 | 0.019 | 0.397 |
| 0.900 | 1 | 0.642 | 0.091 |
| 0.900 | 2 | 0.774 | 0.110 |
| 0.900 | 3 | 0.866 | 0.029 |
| 0.900 | 4 | 0.899 | 0.019 |
| 0.900 | 5 | 1.260 | 0.003 |
| 0.900 | 6 | 1.185 | 0.001 |
| 0.900 | 7 | 1.463 | 0.001 |
| 0.900 | 8 | 1.701 | 0.000 |
| 0.900 | 9 | 1.655 | 0.001 |
| 0.900 | 10 | 1.725 | 0.006 |
| 0.900 | 11 | 1.762 | 0.095 |
| 0.900 | 12 | 2.294 | 0.140 |
| 0.900 | 13 | 2.182 | 0.213 |
| 0.900 | 14 | 2.598 | 0.187 |
| 0.900 | 15 | 2.208 | 0.281 |
| 0.950 | 0 | -0.016 | 0.489 |
| 0.950 | 1 | 0.319 | 0.437 |
| 0.950 | 2 | 0.180 | 0.723 |
| 0.950 | 3 | -0.243 | 0.553 |
| 0.950 | 4 | 0.024 | 0.953 |
| 0.950 | 5 | 0.274 | 0.532 |
| 0.950 | 6 | 0.221 | 0.527 |
| 0.950 | 7 | 0.179 | 0.709 |
| 0.950 | 8 | 0.355 | 0.532 |
| 0.950 | 9 | 0.313 | 0.593 |
| 0.950 | 10 | 0.636 | 0.370 |
| 0.950 | 11 | 1.398 | 0.076 |
| 0.950 | 12 | 2.193 | 0.022 |
| 0.950 | 13 | 2.444 | 0.031 |
| 0.950 | 14 | 2.135 | 0.082 |
| | | Continued on next page | |

| Confidence threshold | Trained task | Accuracy difference | p |
|---:|---:|---:|---:|
| 0.950 | 15 | 2.080 | 0.112 |
| 0.990 | 0 | -0.010 | 0.672 |
| 0.990 | 1 | 0.037 | 0.930 |
| 0.990 | 2 | -0.421 | 0.468 |
| 0.990 | 3 | -0.130 | 0.776 |
| 0.990 | 4 | -0.032 | 0.938 |
| 0.990 | 5 | 0.820 | 0.060 |
| 0.990 | 6 | 0.768 | 0.025 |
| 0.990 | 7 | 0.966 | 0.022 |
| 0.990 | 8 | 1.178 | 0.015 |
| 0.990 | 9 | 1.278 | 0.011 |
| 0.990 | 10 | 1.581 | 0.026 |
| 0.990 | 11 | 1.413 | 0.091 |
| 0.990 | 12 | 2.155 | 0.037 |
| 0.990 | 13 | 2.213 | 0.056 |
| 0.990 | 14 | 2.590 | 0.034 |
| 0.990 | 15 | 2.685 | 0.036 |

Table 5: Results of Student's T-test for RNVP+VCL model configuration.

## A.2 MANN-WHITNEY U TEST

| Confidence threshold | Trained task | Accuracy difference | p |
|---:|---:|---:|---:|
| 0.900 | 0 | -0.006 | 0.877 |
| 0.900 | 1 | 0.303 | 0.248 |
| 0.900 | 2 | 0.453 | 0.424 |
| 0.900 | 3 | 0.539 | 0.732 |
| 0.900 | 4 | 1.852 | 0.024 |
| 0.900 | 5 | 1.600 | 0.109 |
| 0.900 | 6 | 1.231 | 0.306 |
| 0.900 | 7 | 1.990 | 0.059 |
| 0.900 | 8 | 1.994 | 0.082 |
| 0.900 | 9 | 2.431 | 0.031 |
| 0.900 | 10 | 3.564 | 0.001 |
| 0.900 | 11 | 4.040 | 0.000 |
| 0.900 | 12 | 4.323 | 0.000 |
| 0.900 | 13 | 4.884 | 0.000 |
| 0.900 | 14 | 4.816 | 0.000 |
| 0.900 | 15 | 4.928 | 0.000 |
| 0.950 | 0 | 0.027 | 0.304 |
| 0.950 | 1 | 0.059 | 0.880 |
| 0.950 | 2 | 0.545 | 0.284 |
| 0.950 | 3 | 0.921 | 0.213 |
| 0.950 | 4 | 2.388 | 0.002 |
| 0.950 | 5 | 1.744 | 0.036 |
| 0.950 | 6 | 1.312 | 0.134 |
| 0.950 | 7 | 2.145 | 0.053 |
| 0.950 | 8 | 2.373 | 0.030 |
| 0.950 | 9 | 2.132 | 0.027 |
| 0.950 | 10 | 3.047 | 0.002 |
| 0.950 | 11 | 3.258 | 0.000 |
| 0.950 | 12 | 3.257 | 0.000 |
| 0.950 | 13 | 3.636 | 0.000 |
| | | Continued on next page | |

| Confidence threshold | Trained task | Accuracy difference | p |
|---|---|---|---|
| 0.950 | 14 | 3.708 | 0.000 |
| 0.950 | 15 | 4.113 | 0.000 |
| 0.990 | 0 | 0.024 | 0.270 |
| 0.990 | 1 | 0.236 | 0.322 |
| 0.990 | 2 | 0.140 | 0.752 |
| 0.990 | 3 | -0.400 | 0.429 |
| 0.990 | 4 | 0.842 | 0.229 |
| 0.990 | 5 | -0.249 | 0.734 |
| 0.990 | 6 | -1.348 | 0.117 |
| 0.990 | 7 | -0.666 | 0.660 |
| 0.990 | 8 | -0.255 | 0.829 |
| 0.990 | 9 | -0.165 | 0.393 |
| 0.990 | 10 | 0.783 | 0.585 |
| 0.990 | 11 | 1.764 | 0.765 |
| 0.990 | 12 | 2.520 | 0.121 |
| 0.990 | 13 | 3.453 | 0.004 |
| 0.990 | 14 | 3.191 | 0.011 |
| 0.990 | 15 | 3.951 | 0.000 |

Table 6: Results of Mann-Whitney U test for VAE+BNN model configuration.

| Confidence threshold | Trained task | Accuracy difference | p |
|---|---|---|---|
| 0.900 | 0 | 0.030 | 0.631 |
| 0.900 | 1 | -0.107 | 0.767 |
| 0.900 | 2 | -0.240 | 0.714 |
| 0.900 | 3 | -0.503 | 0.439 |
| 0.900 | 4 | 0.422 | 0.094 |
| 0.900 | 5 | 0.678 | 0.139 |
| 0.900 | 6 | 0.839 | 0.181 |
| 0.900 | 7 | 1.912 | 0.012 |
| 0.900 | 8 | 2.054 | 0.016 |
| 0.900 | 9 | 2.611 | 0.005 |
| 0.900 | 10 | 2.777 | 0.006 |
| 0.900 | 11 | 2.629 | 0.010 |
| 0.900 | 12 | 3.122 | 0.007 |
| 0.900 | 13 | 3.684 | 0.004 |
| 0.900 | 14 | 3.888 | 0.004 |
| 0.900 | 15 | 3.762 | 0.009 |
| 0.950 | 0 | 0.170 | 0.006 |
| 0.950 | 1 | -0.460 | 0.038 |
| 0.950 | 2 | -1.810 | 0.000 |
| 0.950 | 3 | -2.529 | 0.000 |
| 0.950 | 4 | -1.653 | 0.005 |
| 0.950 | 5 | -0.999 | 0.036 |
| 0.950 | 6 | -0.765 | 0.092 |
| 0.950 | 7 | 0.267 | 0.980 |
| 0.950 | 8 | 0.740 | 0.675 |
| 0.950 | 9 | 1.529 | 0.156 |
| 0.950 | 10 | 2.026 | 0.133 |
| 0.950 | 11 | 2.399 | 0.020 |
| 0.950 | 12 | 3.422 | 0.001 |
| 0.950 | 13 | 4.488 | 0.000 |
| 0.950 | 14 | 4.380 | 0.000 |

Continued on next page

| Confidence threshold | Trained task | Accuracy difference | p |
|---|---|---|---|
| 0.950 | 15 | 4.056 | 0.002 |
| 0.990 | 0 | 0.018 | 0.875 |
| 0.990 | 1 | -0.090 | 0.855 |
| 0.990 | 2 | -2.402 | 0.001 |
| 0.990 | 3 | -4.201 | 0.000 |
| 0.990 | 4 | -3.287 | 0.000 |
| 0.990 | 5 | -2.986 | 0.000 |
| 0.990 | 6 | -2.263 | 0.001 |
| 0.990 | 7 | -0.966 | 0.125 |
| 0.990 | 8 | -0.732 | 0.132 |
| 0.990 | 9 | 0.983 | 0.693 |
| 0.990 | 10 | 0.913 | 0.982 |
| 0.990 | 11 | 1.599 | 0.457 |
| 0.990 | 12 | 2.730 | 0.115 |
| 0.990 | 13 | 3.939 | 0.068 |
| 0.990 | 14 | 4.530 | 0.035 |
| 0.990 | 15 | 3.998 | 0.108 |

Table 7: Results of Mann-Whitney U test for VAE+VCL model configuration.

| Confidence threshold | Trained task | Accuracy difference | p |
|---|---|---|---|
| 0.900 | 0 | -0.026 | 0.664 |
| 0.900 | 1 | 0.085 | 0.953 |
| 0.900 | 2 | -0.831 | 0.136 |
| 0.900 | 3 | -1.077 | 0.142 |
| 0.900 | 4 | -0.838 | 0.119 |
| 0.900 | 5 | -0.512 | 0.245 |
| 0.900 | 6 | -0.064 | 0.716 |
| 0.900 | 7 | 0.596 | 0.489 |
| 0.900 | 8 | 0.549 | 0.549 |
| 0.900 | 9 | 1.318 | 0.227 |
| 0.900 | 10 | 1.545 | 0.142 |
| 0.900 | 11 | 1.362 | 0.193 |
| 0.900 | 12 | 2.354 | 0.028 |
| 0.900 | 13 | 1.521 | 0.193 |
| 0.900 | 14 | 1.585 | 0.201 |
| 0.900 | 15 | 1.880 | 0.112 |
| 0.950 | 0 | -0.053 | 0.180 |
| 0.950 | 1 | 0.129 | 0.991 |
| 0.950 | 2 | -1.124 | 0.076 |
| 0.950 | 3 | -0.716 | 0.366 |
| 0.950 | 4 | -0.651 | 0.432 |
| 0.950 | 5 | -0.330 | 0.519 |
| 0.950 | 6 | 0.270 | 0.681 |
| 0.950 | 7 | 0.211 | 0.991 |
| 0.950 | 8 | 0.410 | 0.769 |
| 0.950 | 9 | 1.480 | 0.275 |
| 0.950 | 10 | 1.637 | 0.149 |
| 0.950 | 11 | 1.554 | 0.163 |
| 0.950 | 12 | 2.877 | 0.012 |
| 0.950 | 13 | 2.770 | 0.036 |
| 0.950 | 14 | 2.642 | 0.047 |
| 0.950 | 15 | 3.145 | 0.017 |

| Confidence threshold | Trained task | Accuracy difference | p |
|---:|---:|---:|---:|
| 0.990 | 0 | -0.045 | 0.347 |
| 0.990 | 1 | 0.929 | 0.084 |
| 0.990 | 2 | -0.112 | 0.681 |
| 0.990 | 3 | 0.320 | 0.860 |
| 0.990 | 4 | 0.381 | 0.639 |
| 0.990 | 5 | 0.920 | 0.119 |
| 0.990 | 6 | 1.572 | 0.010 |
| 0.990 | 7 | 1.313 | 0.073 |
| 0.990 | 8 | 1.102 | 0.236 |
| 0.990 | 9 | 1.453 | 0.177 |
| 0.990 | 10 | 1.797 | 0.084 |
| 0.990 | 11 | 1.844 | 0.073 |
| 0.990 | 12 | 3.180 | 0.006 |
| 0.990 | 13 | 2.852 | 0.017 |
| 0.990 | 14 | 2.707 | 0.050 |
| 0.990 | 15 | 3.173 | 0.013 |

Table 8: Results of Mann-Whitney U test for RNVP+BNN model configuration.

| Confidence threshold | Trained task | Accuracy difference | p |
|---:|---:|---:|---:|
| 0.900 | 0 | 0.019 | 0.348 |
| 0.900 | 1 | 0.642 | 0.109 |
| 0.900 | 2 | 0.774 | 0.199 |
| 0.900 | 3 | 0.866 | 0.063 |
| 0.900 | 4 | 0.899 | 0.023 |
| 0.900 | 5 | 1.260 | 0.004 |
| 0.900 | 6 | 1.185 | 0.000 |
| 0.900 | 7 | 1.463 | 0.001 |
| 0.900 | 8 | 1.701 | 0.000 |
| 0.900 | 9 | 1.655 | 0.002 |
| 0.900 | 10 | 1.725 | 0.006 |
| 0.900 | 11 | 1.762 | 0.076 |
| 0.900 | 12 | 2.294 | 0.093 |
| 0.900 | 13 | 2.182 | 0.218 |
| 0.900 | 14 | 2.598 | 0.272 |
| 0.900 | 15 | 2.208 | 0.522 |
| 0.950 | 0 | -0.016 | 0.631 |
| 0.950 | 1 | 0.319 | 0.487 |
| 0.950 | 2 | 0.180 | 0.947 |
| 0.950 | 3 | -0.243 | 0.301 |
| 0.950 | 4 | 0.024 | 0.989 |
| 0.950 | 5 | 0.274 | 0.457 |
| 0.950 | 6 | 0.221 | 0.426 |
| 0.950 | 7 | 0.179 | 0.153 |
| 0.950 | 8 | 0.355 | 0.060 |
| 0.950 | 9 | 0.313 | 0.152 |
| 0.950 | 10 | 0.636 | 0.113 |
| 0.950 | 11 | 1.398 | 0.048 |
| 0.950 | 12 | 2.193 | 0.022 |
| 0.950 | 13 | 2.444 | 0.033 |
| 0.950 | 14 | 2.135 | 0.118 |
| 0.950 | 15 | 2.080 | 0.164 |
| 0.990 | 0 | -0.010 | 0.815 |

| Confidence threshold | Trained task | Accuracy difference | p |
|---|---|---|---|
| 0.990 | 1 | 0.037 | 0.755 |
| 0.990 | 2 | -0.421 | 0.741 |
| 0.990 | 3 | -0.130 | 0.961 |
| 0.990 | 4 | -0.032 | 0.728 |
| 0.990 | 5 | 0.820 | 0.028 |
| 0.990 | 6 | 0.768 | 0.012 |
| 0.990 | 7 | 0.966 | 0.006 |
| 0.990 | 8 | 1.178 | 0.006 |
| 0.990 | 9 | 1.278 | 0.006 |
| 0.990 | 10 | 1.581 | 0.017 |
| 0.990 | 11 | 1.413 | 0.037 |
| 0.990 | 12 | 2.155 | 0.035 |
| 0.990 | 13 | 2.213 | 0.069 |
| 0.990 | 14 | 2.590 | 0.045 |
| 0.990 | 15 | 2.685 | 0.046 |

Table 9: Results of Mann-Whitney U test for RNVP+VCL model configuration.

## A.3 Mood's test

| Confidence threshold | Trained task | Accuracy difference | p |
|---|---|---|---|
| 0.900 | 0 | 0.010 | 0.617 |
| 0.900 | 1 | 0.225 | 0.453 |
| 0.900 | 2 | -0.003 | 1.000 |
| 0.900 | 3 | -0.430 | 0.803 |
| 0.900 | 4 | 1.258 | 0.211 |
| 0.900 | 5 | 0.472 | 0.901 |
| 0.900 | 6 | 0.549 | 0.530 |
| 0.900 | 7 | 1.509 | 0.530 |
| 0.900 | 8 | 1.670 | 0.096 |
| 0.900 | 9 | 1.032 | 0.071 |
| 0.900 | 10 | 1.683 | 0.018 |
| 0.900 | 11 | 1.932 | 0.000 |
| 0.900 | 12 | 2.789 | 0.000 |
| 0.900 | 13 | 3.458 | 0.000 |
| 0.900 | 14 | 3.516 | 0.000 |
| 0.900 | 15 | 3.055 | 0.000 |
| 0.950 | 0 | 0.040 | 0.080 |
| 0.950 | 1 | 0.075 | 0.901 |
| 0.950 | 2 | 0.368 | 0.901 |
| 0.950 | 3 | 0.719 | 0.102 |
| 0.950 | 4 | 2.331 | 0.008 |
| 0.950 | 5 | 1.423 | 0.102 |
| 0.950 | 6 | 1.006 | 0.530 |
| 0.950 | 7 | 1.574 | 0.204 |
| 0.950 | 8 | 2.041 | 0.091 |
| 0.950 | 9 | 1.553 | 0.242 |
| 0.950 | 10 | 1.353 | 0.086 |
| 0.950 | 11 | 1.919 | 0.003 |
| 0.950 | 12 | 2.307 | 0.003 |
| 0.950 | 13 | 2.714 | 0.000 |
| 0.950 | 14 | 2.863 | 0.000 |
| 0.950 | 15 | 2.752 | 0.000 |
| | | Continued on next page | |

| Confidence threshold | Trained task | Accuracy difference | p |
|---:|---:|---:|---:|
| 0.990 | 0 | 0.040 | 0.080 |
| 0.990 | 1 | 0.130 | 0.901 |
| 0.990 | 2 | -0.265 | 1.000 |
| 0.990 | 3 | -1.064 | 0.377 |
| 0.990 | 4 | 0.747 | 0.901 |
| 0.990 | 5 | -0.672 | 0.366 |
| 0.990 | 6 | -1.542 | 0.157 |
| 0.990 | 7 | -0.164 | 1.000 |
| 0.990 | 8 | 0.523 | 0.900 |
| 0.990 | 9 | -0.276 | 0.686 |
| 0.990 | 10 | -0.951 | 0.715 |
| 0.990 | 11 | -0.032 | 1.000 |
| 0.990 | 12 | 0.683 | 0.233 |
| 0.990 | 13 | 1.501 | 0.047 |
| 0.990 | 14 | 1.702 | 0.233 |
| 0.990 | 15 | 1.726 | 0.005 |

Table 10: Results of Mood's test for VAE+BNN model configuration.

| Confidence threshold | Trained task | Accuracy difference | p |
|---:|---:|---:|---:|
| 0.900 | 0 | 0.060 | 0.527 |
| 0.900 | 1 | 0.465 | 0.639 |
| 0.900 | 2 | 0.642 | 0.266 |
| 0.900 | 3 | 0.054 | 1.000 |
| 0.900 | 4 | 1.235 | 0.079 |
| 0.900 | 5 | 1.460 | 0.079 |
| 0.900 | 6 | 1.601 | 0.266 |
| 0.900 | 7 | 2.726 | 0.079 |
| 0.900 | 8 | 2.715 | 0.104 |
| 0.900 | 9 | 3.011 | 0.104 |
| 0.900 | 10 | 2.794 | 0.023 |
| 0.900 | 11 | 2.380 | 0.023 |
| 0.900 | 12 | 2.438 | 0.023 |
| 0.900 | 13 | 2.956 | 0.104 |
| 0.900 | 14 | 3.653 | 0.071 |
| 0.900 | 15 | 3.135 | 0.251 |
| 0.950 | 0 | 0.145 | 0.093 |
| 0.950 | 1 | -0.568 | 0.171 |
| 0.950 | 2 | -2.365 | 0.010 |
| 0.950 | 3 | -2.864 | 0.000 |
| 0.950 | 4 | -1.587 | 0.010 |
| 0.950 | 5 | -1.083 | 0.064 |
| 0.950 | 6 | -0.529 | 0.217 |
| 0.950 | 7 | 0.511 | 1.000 |
| 0.950 | 8 | 0.944 | 0.343 |
| 0.950 | 9 | 1.376 | 0.114 |
| 0.950 | 10 | 1.734 | 0.425 |
| 0.950 | 11 | 2.213 | 0.038 |
| 0.950 | 12 | 2.538 | 0.007 |
| 0.950 | 13 | 3.876 | 0.001 |
| 0.950 | 14 | 3.707 | 0.001 |
| 0.950 | 15 | 3.153 | 0.009 |
| 0.990 | 0 | 0.080 | 0.266 |

| Confidence threshold | Trained task | Accuracy difference | p |
|---|---|---|---|
| 0.990 | 1 | 0.098 | 1.000 |
| 0.990 | 2 | -1.360 | 0.036 |
| 0.990 | 3 | -3.144 | 0.000 |
| 0.990 | 4 | -2.494 | 0.000 |
| 0.990 | 5 | -2.523 | 0.000 |
| 0.990 | 6 | -1.903 | 0.006 |
| 0.990 | 7 | -0.703 | 0.863 |
| 0.990 | 8 | -1.398 | 0.330 |
| 0.990 | 9 | 0.670 | 0.745 |
| 0.990 | 10 | 0.235 | 1.000 |
| 0.990 | 11 | 0.622 | 0.642 |
| 0.990 | 12 | 1.668 | 0.100 |
| 0.990 | 13 | 3.004 | 0.100 |
| 0.990 | 14 | 3.763 | 0.081 |
| 0.990 | 15 | 3.123 | 0.214 |

Table 11: Results of Mood's test for VAE+VCL model configuration.

| Confidence threshold | Trained task | Accuracy difference | p |
|---|---|---|---|
| 0.900 | 0 | -0.010 | 1.000 |
| 0.900 | 1 | 0.323 | 0.763 |
| 0.900 | 2 | -1.393 | 0.132 |
| 0.900 | 3 | -1.400 | 0.132 |
| 0.900 | 4 | -1.284 | 0.132 |
| 0.900 | 5 | -1.151 | 0.132 |
| 0.900 | 6 | -0.091 | 1.000 |
| 0.900 | 7 | 0.572 | 0.366 |
| 0.900 | 8 | 1.067 | 0.366 |
| 0.900 | 9 | 2.141 | 0.366 |
| 0.900 | 10 | 2.065 | 0.132 |
| 0.900 | 11 | 1.845 | 0.132 |
| 0.900 | 12 | 2.954 | 0.132 |
| 0.900 | 13 | 2.865 | 0.763 |
| 0.900 | 14 | 2.541 | 0.366 |
| 0.900 | 15 | 2.711 | 0.448 |
| 0.950 | 0 | -0.035 | 0.762 |
| 0.950 | 1 | -0.200 | 0.366 |
| 0.950 | 2 | -1.428 | 0.132 |
| 0.950 | 3 | -0.722 | 0.763 |
| 0.950 | 4 | -0.528 | 0.763 |
| 0.950 | 5 | -0.705 | 0.366 |
| 0.950 | 6 | 0.447 | 0.763 |
| 0.950 | 7 | -0.094 | 1.000 |
| 0.950 | 8 | 0.606 | 0.763 |
| 0.950 | 9 | 1.468 | 0.366 |
| 0.950 | 10 | 1.778 | 0.366 |
| 0.950 | 11 | 1.161 | 0.132 |
| 0.950 | 12 | 2.500 | 0.132 |
| 0.950 | 13 | 2.979 | 0.763 |
| 0.950 | 14 | 2.494 | 0.763 |
| 0.950 | 15 | 3.145 | 0.448 |
| 0.990 | 0 | -0.080 | 0.366 |
| 0.990 | 1 | 0.940 | 0.132 |

Continued on next page

| Confidence threshold | Trained task | Accuracy difference | p |
|---|---|---|---|
| 0.990 | 2 | -0.488 | 0.763 |
| 0.990 | 3 | -0.451 | 0.763 |
| 0.990 | 4 | 0.203 | 1.000 |
| 0.990 | 5 | 0.773 | 0.035 |
| 0.990 | 6 | 1.453 | 0.007 |
| 0.990 | 7 | 1.046 | 0.132 |
| 0.990 | 8 | 0.570 | 0.132 |
| 0.990 | 9 | 2.335 | 0.366 |
| 0.990 | 10 | 2.104 | 0.366 |
| 0.990 | 11 | 1.387 | 0.035 |
| 0.990 | 12 | 1.982 | 0.048 |
| 0.990 | 13 | 3.269 | 0.171 |
| 0.990 | 14 | 2.216 | 0.448 |
| 0.990 | 15 | 3.003 | 0.171 |

Table 12: Results of Mood's test for RNVP+BNN model configuration.

| Confidence threshold | Trained task | Accuracy difference | p |
|---|---|---|---|
| 0.900 | 0 | 0.025 | 0.425 |
| 0.900 | 1 | 0.450 | 0.037 |
| 0.900 | 2 | 0.513 | 0.233 |
| 0.900 | 3 | 0.483 | 0.454 |
| 0.900 | 4 | 0.516 | 0.233 |
| 0.900 | 5 | 1.111 | 0.037 |
| 0.900 | 6 | 1.451 | 0.037 |
| 0.900 | 7 | 1.909 | 0.003 |
| 0.900 | 8 | 2.186 | 0.011 |
| 0.900 | 9 | 1.852 | 0.043 |
| 0.900 | 10 | 1.649 | 0.015 |
| 0.900 | 11 | 2.645 | 0.114 |
| 0.900 | 12 | 2.239 | 0.155 |
| 0.900 | 13 | 1.480 | 0.155 |
| 0.900 | 14 | 2.298 | 0.155 |
| 0.900 | 15 | 2.888 | 0.653 |
| 0.950 | 0 | 0.000 | 0.924 |
| 0.950 | 1 | 0.252 | 0.175 |
| 0.950 | 2 | -0.028 | 1.000 |
| 0.950 | 3 | -0.404 | 0.303 |
| 0.950 | 4 | 0.034 | 1.000 |
| 0.950 | 5 | 0.698 | 0.233 |
| 0.950 | 6 | 0.474 | 0.454 |
| 0.950 | 7 | 0.948 | 0.101 |
| 0.950 | 8 | 1.259 | 0.036 |
| 0.950 | 9 | 0.852 | 0.261 |
| 0.950 | 10 | 1.504 | 0.103 |
| 0.950 | 11 | 1.767 | 0.049 |
| 0.950 | 12 | 1.184 | 0.005 |
| 0.950 | 13 | 1.934 | 0.106 |
| 0.950 | 14 | 0.880 | 0.106 |
| 0.950 | 15 | 1.727 | 0.465 |
| 0.990 | 0 | 0.030 | 0.281 |
| 0.990 | 1 | 0.092 | 0.761 |
| 0.990 | 2 | -0.227 | 0.888 |

| Confidence threshold | Trained task | Accuracy difference | p |
|---|---|---|---|
| 0.990 | 3 | -0.075 | 1.000 |
| 0.990 | 4 | 0.296 | 0.761 |
| 0.990 | 5 | 0.929 | 0.037 |
| 0.990 | 6 | 0.861 | 0.037 |
| 0.990 | 7 | 0.862 | 0.101 |
| 0.990 | 8 | 1.201 | 0.099 |
| 0.990 | 9 | 1.248 | 0.043 |
| 0.990 | 10 | 1.756 | 0.118 |
| 0.990 | 11 | 2.259 | 0.159 |
| 0.990 | 12 | 2.486 | 0.116 |
| 0.990 | 13 | 2.706 | 0.322 |
| 0.990 | 14 | 3.508 | 0.116 |
| 0.990 | 15 | 3.600 | 0.365 |

Table 13: Results of Mood's test for RNVP+VCL model configuration.

