# OpenReview forum: "Self-Supervised Pseudodata Filtering for Improved Replay with Sub-Optimal Generators"
_ICLR.cc/2024/Conference — Submitted to ICLR 2024_

### Official Review · Reviewer_6px8 · 2023-10-27

**Soundness:** 2 fair
**Presentation:** 3 good
**Contribution:** 1 poor
**Rating:** 1
**Confidence:** 5

**Summary:**

The paper addresses class incremental learning (CIL) by proposing an improvement to generative replay strategies. This strategy requires continual updating of both a classifier, and a generative model. The purpose of the generative model is to act like a memory for data from past tasks, for which the original training data is no longer available. One known issue with training the generative model is that it can degrade due to being forced to continually update itself using data it generates itself.

This paper proposes a way to improve the generative model within the generative replay strategy. The contribution of the paper is to propose a filter that rejects or accepts generated samples, based on whether the classifier can correctly classify the generated samples with high confidence.

**Strengths:**

Strengths of the paper:

- Originality: The paper identifies a central weakness is past generate replay strategies, and proposes a way to overcome that weakness. The idea of using the generator and classifier within generative replay CIL is reminiscent of how GANs are trained, which is a neat idea.
- Clarity: The writing is very clear, and it was easy to follow in all sections. The results are well presented, and thorough in their statistical rigor.
- Significance: The proposed method is potentially important for research in CIL, in which there is renewed interest in generative models. The  coverage of multiple kinds of generative models looked at here is a likely direction for the field, and the central idea of the paper is certainly  applicable to any generative model in the CIL context.

**Weaknesses:**

The paper has a clear overarching weakness which is its lack of testing of the method on meaningful standard CIL datasets. The only dataset considered is EMNIST. Although EMNIST is marginally more challenging than MNIST, I am of the view that its not possible to push forward the CIL field by testing a new approach only on near binary handwritten characters. Nowadays, every new paper on CIL in high profile conferences uses challenging colour image datasets such as CIFAR100, Tiny Imagenet etc, at the very least.

For this reviewer, a minimal precondition in revising this paper will be to show results on at least three datasets, including those for which the literature lists comparison results. The more common CIL datasets for which it's easy to find comparison results in the recent literature (e.g. 2023, 2022) are split CIFAR 100 and split tiny Imagenet, both with e.g. 5, 10 or 20 CIL tasks.  If the authors can show that their generative replay filter provides a benefit on these datasets within the same framework, then I will be willing to reconsider my recommendation.

The paper also does not sufficiently review related work in the continual learning literature. The only generative replay papers references are from 2021 or earlier.  A quick search for work from the last two years found several relevant works that cover basically the same ground as this paper, but test their methods in comparison with competing methods on datasets like split CIFAR-100:

1. @InProceedings{Khan_2023_ICCV,
    author    = {Khan, Valeriya and Cygert, Sebastian and Twardowski, Bartlomiej and Trzci\'nski, Tomasz},
    title     = {Looking Through the Past: Better Knowledge Retention for Generative Replay in Continual Learning},
    booktitle = {Proceedings of the IEEE/CVF International Conference on Computer Vision (ICCV) Workshops},
    month     = {October},
    year      = {2023},
    pages     = {3496-3500}
}

2. @inproceedings{10.5555/3618408.3618842,
author = {Gao, Rui and Liu, Weiwei},
title = {DDGR: Continual Learning with Deep Diffusion-Based Generative Replay},
year = {2023},
publisher = {JMLR.org},
abstract = {Popular deep-learning models in the field of image classification suffer from catastrophic forgetting--models will forget previously acquired skills when learning new ones. Generative replay (GR), which typically consists of a generator and a classifier, is an efficient way to mitigate catastrophic forgetting. However, conventional GR methods only focus on a single instruction relationship (generator-to-classifier), where the generator synthesizes samples for previous tasks to instruct the training of the classifier, while ignoring the ways in which the classifier can benefit the generator. In addition, most generative replay methods typically reuse the generated samples to update the generator, which causes the samples regenerated by the generator deviating from the distribution of previous tasks. To overcome these two issues, we propose a novel approach, called deep diffusion-based generative replay (DDGR), which adopts a diffusion model as the generator and calculates an instruction-operator through the classifier to instruct the generation of samples. Extensive experiments in class incremental (CI) and class incremental with repetition (CIR) settings demonstrate the advantages of DDGR. Our code is available at https://github.com/xiaocangshengGR/DDGR.},
booktitle = {Proceedings of the 40th International Conference on Machine Learning},
articleno = {434},
numpages = {20},
location = {Honolulu, Hawaii, USA},
series = {ICML'23}
}


Minor comment: in the first para, it is stated that "Class-IL model lacks such knowledge altogether". This is not correct. For CIL, the task boundaries are known during training.

**Questions:**

The result all seem to show accuracy differences. What were the actual accuracies? Without knowing these and how they compare with competing CIL methods, it's unclear if the accuracy improvement is of any value at all.

---

### Official Review · Reviewer_yxwt · 2023-11-01

**Soundness:** 2 fair
**Presentation:** 2 fair
**Contribution:** 2 fair
**Rating:** 3
**Confidence:** 5

**Summary:**

The paper addresses the challenge of continual learning in deep neural networks, particularly in a class-incremental scenario where new classes are introduced over time. To tackle this, the authors propose a filtering mechanism that allows the classifier to reject low-confidence samples when generating pseudodata for new tasks. Experimental results verify the effectiveness of the method.

**Strengths:**

(1) Generative replay for continual learning is an interesting direction.

**Weaknesses:**

(1) **The paper's citation count is minimal, and it lacks a dedicated *Related Work* section.** Considering the paper's focus on sample replay in continual learning, it is essential to include discussion and citations related to this topic, particularly concerning generative sample replay.

(2) **The paper lacks a comparison with existing methods for continual learning, and the entire experimental section is rather perplexing.** It is advisable for the paper to present experimental findings using well-established datasets such as CIFAR-100 and ImageNet, along with comparisons to other methodologies. For instance, it could include a comparison with a study that similarly employs the concept of replaying synthetic images for continual learning, as done in the work by [Smith et al., 2021].

***References:***

[Smith et al., 2021] James Smith, Yen-Chang Hsu, Jonathan Balloch, Yilin Shen, Hongxia Jin, Zsolt Kira. Always Be Dreaming: A New Approach for Data-Free Class-Incremental Learning. ICCV 2021.

**Questions:**

Please refer to [Weaknesses].

---

### Official Review · Reviewer_fB4Z · 2023-11-11

**Soundness:** 2 fair
**Presentation:** 2 fair
**Contribution:** 1 poor
**Rating:** 3
**Confidence:** 4

**Summary:**

The authors introduce a variation on generative replay which rejection-samples examples the classifier has insufficiently high confidence on. This is a general-purpose change which can be made on top of any continual learning model. The authors experiment with a variety of models in the setting of class-incremental learning and show the technique gives modest improvements as the number of tasks added increases.

**Strengths:**

There is ample contextualization and motivation of the work.
The experiments across architectures are thorough and represent an interesting set of architectures.
There is some good discussion of generalization and why the method seems to hurt at lower numbers of tasks.

**Weaknesses:**

Catastrophic forgetting absolutely does affect biological learning agents, just not in the same way or to the same degree maybe; I suggest rephrasing this. Also, biological experience replay is a hypothesis; it's certainly a reasonable one but it's presented as "responsible for this evolutionary success" which is not a very good/accurate way to describe a scientific hypothesis. IMO you could just remove this (second p in intro) paragraph , it doesn't add anything relevant to the paper.

The text is a bit convolutedly-written, making it not very clear to read. E.g. "If pseudodata generated for one of the tasks contains features unnecessary or confusing for the classifer, there exists a chance that these features are going to be preserved in the distribution learned by the generator, detrimentally affecting replay’s effectiveness for all the subsequent tasks"

Differentiation from Aljundi et al is not clear "the policy we propose can be treated as a stricter variant, when pseudodata quality is assessed on the dataset level instead of just encouraging the model to improve it with time" -- I get it, based on reading further, but this sentence is not giving a clear description, and it also doesn't give a reason we should expect the proposed approach to be significantly better or more well-motivated in any way (maybe it's computationally more efficient?). Because it's such a similar method, there should be a comparison run to see if the results differ significantly.

In general, there is not much related work directly similar to the proposed technique. I'm not an expert in this area, but rejection sampling is a well-known general technique, I would be surprised if it has not been tried in other continual learning settings. E.g. I did a quick search and found this paper which does this for planning in robotics, and it references two other papers doing the same: https://arxiv.org/pdf/2208.07737.pdf (rejection sampling is only a component of what they're doing, maybe there are more directly similar works I'm unaware of).

There's a lot of space taken up explaining all the models. A big point of your method is that it's general-purpose, so the exact model doesn't really matter. It's great that you tried a variety of models, but the level of detail here belongs in appendix in my opinion, except for any specific facts about the reasons you chose each model that are directly relevant to the point of your paper. Even in appendix, you do not need (should not have?) multiple paragraphs to explain real NVP, VAEs, and the variational lower bound. This is not an essay. Thwe whole point of citing prior literature in scientific work is so that you don't have to exhaustively reexplain it. Only pull out details that you're changing/adapting to your context -- e.g.  "The VAE used in this work was implemented as a convolutional, non-conditional generator – a
single, two-dimensional latent distribution was shared between classes." But also, why these choices? Use the space to explain your experimental choices, not the previously-published architecture.

The Training Procedure figure 2 is not well designed. It's hard to read with all the different text sizes, and takes up a very large amount of unnecessary vertical space. It is unclear both in the figure and in the caption what are the conditions to go from one stage to another, and how long we stay in each loop. It's not necessary to have both this figure and the pseudocode; one could go in the appendix.

**Questions:**

Overall, you've done some good work here, on an interesting problem, but it would need a lot of work in my opinion to meet the bar. The experiments are not adequate to establish significant difference from prior work, and there  isn't enough discussion of prior work directly relevant to the actual novelty proposed in the paper (i.e. rejection sampling and generalization, e.g. in GANS and in continual learning).

Rejection sampling based on classifier confidence and then evaluating that technique based on the same classifier's performance seems deeply inadequate to me -- it's sort of just making the classifier more confident, without necessarily improving it at all. I would want to see alternative evaluation methods that demonstrate the method is actually doing something useful. This could fit well with a more in-depth analysis of the scaling with tasks, and alternative tasks -- Why focus on only class-incremental learning?


minor
- I suggest doing an edit pass for clarity on the text (after moving substantial portions to appendix to make space for more experiments).
 - class-incremental is a type of open-set learning, suggest adding "aka open-set learning" in abstract for readers who might be more familiar with that term.
 - things are a bit overstated in the abs overall, e.g. the model doesnt "have to" revisit past classes,  the generator doesn't "need" to learn continually; those are just common approaches
- quotes messed up around replay and rehearsal
 - "deeming" -> making, or rendering
 - pseuodata used without that term being defined/explained
 - the results figures are not colour-blind friendly, labels are small and hard to read, and the surface area of the plots is unnecessarily large. Captions should include interpretation of the results.

---

### Author Response · Authors · 2023-11-22

Dear Reviewers,

Thank you for your comprehensive and valuable feedback. Further experimental investigation on more advanced datasets is indeed necessary, but preparing and performing the tests require more time than this discussion period allows. Your remarks regarding discussing the related work and general editorial shortcomings will also be addressed before an eventual resubmission. Two of your questions can be however answered now:

1. Reviewer **fB4Z**'s question regarding the evaluation method: Please consider that while rejection sampling is based on the classifier's confidence regarding the generated pseudodata, these samples are not used to evaluate the model's performance. The reported accuracy differences were calculated on a separate test dataset. This set contained exclusively real data which was never presented to the classifier or the generator during training. Given that, in our interpretation, this method does not simply "make the classifier more confident", since the classifier gets no feedback about its performance on the test set. Nevertheless, alternative methods of evaluation would be definitely beneficial to the article.
2. Reviewer **6px8**'s question regarding the accuracies: Including the absolute values of the metrics, at least in the appendix, would indeed make the paper more transparent. We decided to focus on the differences while reporting the results because our goal was to show a *relative* improvement when using a *sub-optimal* generator, and how it scales with the size of the problem. If the goal was to reach state-of-the-art absolute performance, then the biggest improvement in our framework would be probably achieved by optimizing the generator's hyperparameters. Instead, we required the generator to be just "sufficiently good" to perform the experiments.

Thank you once again for the time and effort you put in reviewing of this manuscript.

---

### Meta-Review · Area_Chair_qJkd · 2023-12-05

**Metareview:**

The paper tackles the challenge of avoiding catastrophic interference in sequential/continual learning scenarios from an angle of augmenting replay strategies. Although the reviewers all agree that there are strengths in the motivation of the work, there is consensus that the current execution, in particular with respect to related work and respective empirical comparison needs a revisit before publication. In the discussion process, the authors have acknowledged the constructive feedback and seem to further agree that more time will be required to properly include revisions. A resubmission will be required.

**Justification For Why Not Higher Score:**

The suggested changes require more time to be implemented and will require another round of peer review before the paper an be published. This seems to be in accord with the authors themselves, who state that revisions will be accounted for prior to resubmission.

**Justification For Why Not Lower Score:**

N/A

---

### Decision · Program_Chairs · 2024-01-16

Reject